# Neutrophil-to-lymphocyte ratio associated with an increased risk of mortality in patients with critical limb ischemia

**Min-I. Su[1,2], Cheng-Wei Liu**[3,4]*

**1** Division of Cardiology, Department of Internal Medicine, Taitung MacKay Memorial Hospital, Taitung City, Taiwan, **2** Department of Medicine, MacKay Medical College, New Taipei City, Taiwan, **3** Department of Internal Medicine, Tri-Service General Hospital Songshan Branch, National Defense Medical Center, Taipei, Taiwan, **4** Graduate Institute of Clinical Medicine, College of Medicine, National Taiwan University, Taipei, Taiwan

* issac700319@gmai.com

**Data Availability Statement:** All relevant data are within the manuscript and its Supporting information files.

**Funding:** The authors received no specific funding for this work.

## Abstract

### Purpose

Association of the neutrophil-to-lymphocyte ratio (NLR) with mortality has not been comprehensively explored in critical limb ischemia (CLI) patients. We investigated the association between the NLR and clinical outcomes in CLI.

### Materials and methods

We retrospectively enrolled consecutive CLI patients between 1/1/2013 and 12/31/2018. Receiver operating characteristic curve analysis determined NLR cutoffs for 1-year in-hospital, all-cause and cardiac-related mortality; major adverse cardiovascular events (MACEs); and major adverse limb events (MALEs).

### Results

Among 195 patients (age, 74.0 years, SD: 11.5; 51.8% male; body mass index, 23.4 kg/m$^2$, SD: 4.2), 14.4% exhibited acute limb ischemia. After 1 year, patients with NLR>8 had higher in-hospital mortality (21.1% vs. 3.6%, P<0.001), all-cause mortality (54.4% vs. 13.8%, P<0.001), cardiac-related mortality (28.1% vs. 6.5%, P<0.001), MACE (29.8% vs. 13.0%, P = 0.008), and MALE (28.1% vs. 13.0%, P = 0.021) rates than those with NLR<8. In multivariate logistic regression, NLR≥8 was significantly associated with all-cause (P<0.001) and cardiac-related (adjusted HR: 5.286, 95% CI: 2.075–13.47, P<0.001) mortality, and NLR≥6 was significantly associated with MALEs (adjusted HR: 2.804, 95% CI: 1.292–6.088, P = 0.009). Each increase in the NLR was associated with increases in all-cause (adjusted HR: 1.028, 95% CI: 1.008–1.049, P = 0.007) and cardiac-related (adjusted HR:1.027, 95% CI: 0.998–1.057, P = 0.073) mortality but not in-hospital mortality or MACEs.

**Competing interests:** The authors have declared
that no competing interests exist.

## Conclusion

CLI patients with high NLRs had significantly higher risks of 1-year all-cause and cardiac-related mortality and MALEs. The NLR can be used for prognostic prediction in these patients.

## Introduction

The neutrophil-to-lymphocyte ratio (NLR) is widely used as a prognostic biomarker in various diseases, such as cancer and cardiovascular disease [1, 2]. Both of these disorders have a common pathophysiology involving inflammatory processes that can be roughly represented as the ratio of neutrophils [3, 4]; the proportion of lymphocytes indicates the host immune response and has been associated with mortality in healthy individuals [5]. The NLR combines the properties of the inflammatory and immune responses and thereby enables the prediction of outcomes in patients with diverse atherosclerotic cardiovascular and peripheral vascular diseases [6, 7].

An elevated NLR has been associated with unfavorable neurological outcomes and increased mortality in patients with ischemic stroke [8], with an increased risk of mortality and major adverse cardiovascular events (MACE) in patients with acute myocardial infarction [9], and with the severity of lower extremity artery disease (LEAD) in cohort studies [10, 11]. Other cohort studies have further reported the association between NLR and mortality in patients with critical limb ischemia (CLI) [12, 13]. However, no studies have reported the comprehensive outcomes of all-cause and cardiac-related mortality, MACE, and major adverse limb events (MALE) in patients with CLI. Therefore, we conducted the present study to investigate the association between the NLR and outcomes in patients with CLI.

## Materials and methods

We retrospectively and continuously enrolled patients with CLI undergoing percutaneous transluminal angioplasty at our hospital between 2013/1/1 and 2018/12/31. The study patients were all-comers, with the only specific exclusion criterion CLI patients with a nonsalvageable limb who refused amputation surgery. We divided the study patients into higher and lower NLR groups and collected the patients' baseline characteristics, laboratory data, procedural details, and outcomes from medical records. All patients were followed up until 2019/12/31. Given that the present study was a retrospective cohort study with a low risk, no informed consent was needed from the study patients. The study was approved by the MacKay Memorial Hospital with Institutional Review Board number (20MMHIS034e).

Patients who present to our emergency department with CLI routinely receive dual antiplatelet therapy with aspirin plus clopidogrel, and heparinization is loaded according to the guidelines unless contraindicated. We do not routinely prescribe cilostazol to patients with CLI. We measured complete blood count and white blood cell differential at our emergency department before the patients received percutaneous transluminal angioplasty. During this study, patients who presented with acute limb ischemia were treated by emergent percutaneous transluminal angioplasty, and patients with chronic limb ischemia were treated by urgent percutaneous transluminal angioplasty. In our cardiac catheterization laboratory, the physicians decided whether to use an antegrade or retrograde approach depending on a patient's lesion type. Heparin was administered to maintain an active clotting time between 250 and

300 seconds. Because of our national health insurance rules, we generally treat iliac and femor-opopliteal lesions with balloon angioplasty, a drug-coated balloon or a stent and below-knee lesions with only balloon angioplasty; use of a drug-coated balloon or stent was left to the physicians' decision and was mainly influenced by the patients' economic status. At discharge, dual anti-platelet agents were continued for at least one month unless clinically significant bleeding complications developed.

CLI was defined according to the Rutherford classification, including rest pain (stage IV), tissue loss (stage V), and gangrene (stage VI) [14]. The primary study outcomes were all-cause mortality, cardiac-related mortality, MACE, and MALE at the one-year follow-up. MACE was defined as the composite of nonfatal myocardial infarction, nonfatal stroke, and cardiac-related death; MALE was defined as an amputation due to a vascular event above the forefoot, acute limb ischemia and clinically driven target vessel revascularization. Secondary outcomes were in-hospital all-cause mortality.

Continuous variables are presented as numbers and standard deviations, and binary variables are presented as numbers and percentages; independent t-tests and chi-squared tests were used to evaluate differences in continuous variables and binary variables, respectively. We used receiver operating characteristic curves to identify NLR cut-off values for each study outcome. We used univariate logistic regression analyses to investigate the associations between NLR and study outcomes and between variables and study outcomes. Variables that were significantly associated with the study outcomes were considered confounders, and they were adjusted in multivariate logistic regression analyses. The presence of acute limb ischemia and Rutherford classification criteria were major risk factors associated with outcomes in patients with CLI and were adjusted in the multivariate logistic regression analyses to determine whether they were significantly associated with the study outcomes identified in the univariate logistic regression analyses. All P values are two-tailed, and P values of 0.05 or lower were considered significant. We performed all statistical analyses with Statistical Package for the Social Sciences (SPSS software, version 20.0).

## Results

The study consisted of 195 patients with CLI who underwent percutaneous transluminal angioplasty after we excluded three patients with a nonsalvageable limb who refused amputation surgery and two patients with missing data for NLR. The patients had a mean age of 74 years, (SD:12) and a mean NLR of 8.2 (SD:10.0), with 52.3% male and Rutherford stages IV, V, and VI accounting for 27.4%, 66.0%, and 6.6% of the patients, respectively. We used receiver operating characteristic curves to identify cut-off values for NLR; these cut-off values were eight for one-year all-cause and cardiac-related mortality and MACE, six for MALE, and five for in-hospital mortality. Fig 1 shows the area under the curve for the study outcomes. The incidences of the primary outcomes were 25.4% for all-cause mortality, 12.7% for cardiac-related mortality, 17.8% for MACE, and 17.3% for MALE, and the incidence of the secondary outcome was 8.6% for in-hospital all-cause mortality. Regarding the Rutherford classification, in CLI patients with Rutherford stages IV, V, and VI, the incidences of all-cause mortality were 18.9%, 26.4%, and 46.2%, respectively, and those of MALE were 0%, 22.5% and 38.5%, respectively.

With regard for the primary study outcomes, there was no significant difference in baseline characteristics, such as the ratio of smoking and chronic kidney disease, a history of amputation, and heart failure status, between patients with NLR ≥ 8 vs. NLR < 8; however, compared with patients with NLR < 8, those with NLR ≥ 8 had higher heart rates (93.3 vs. 86.5 beats per minute, P = 0.014), a higher rate of presenting with acute limb ischemia (29.8% vs. 9.2%,

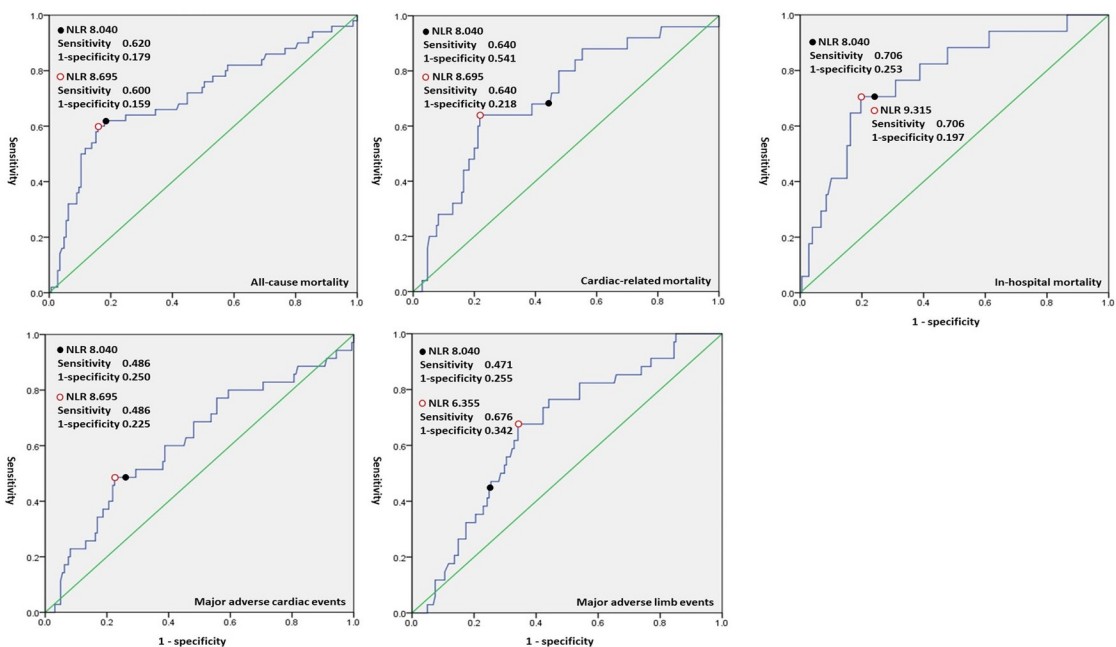

**Fig 1. Receiver operating characteristic curves showing the NLR cut-off for each study outcome.**

P = 0.001) and a more severe Rutherford stage (70.2% vs. 65.2% for stage V, 14.0% vs 3.5% for stage VI, P = 0.005). Medication use at baseline did not differ between the two groups except that there was a lower rate of cilostazol use in the patients with NLR > 8 than in those with NLR < 8 (31.6% vs. 51.8%, P = 0.012). We show the patients' baseline characteristics, laboratory data, and medication use by NLR ≥ 8 vs. < 8 in relation to one-year all-cause and cardiac-related mortality and MACE in Table 1; by NLR ≥ 5 vs. NLR < 5 in relation to in-hospital mortality in S1 Table; and by NLR ≥ 6 vs. < 6 in relation to MALE in S2 Table. In our study, the mean NLR value was 6.1 (SD:7.4) for Rutherford stage IV, 8.6 (SD:1.07) for Rutherford stage V, and 13.2 (SD: 10.6) for Rutherford stage VI.

The primary outcomes were significantly worse in patients with NLR ≥ 8 than in those with NLR < 8 (54.4% vs. 13.8%, P < 0.001 for one-year all-cause mortality; 28.1% vs. 6.5%, P < 0.001 for cardiac-related mortality; 29.8% vs. 13.0%, P = 0.008 for MACE; and 28.1% vs. 13.0%, P = 0.021 for MALE); the secondary outcomes were also significantly worse in patients with NLR ≥ 8 than in the patients with NLR < 8 (21.1% vs. 3.6%, P < 0.001). Regarding the primary outcomes in univariate logistic regression analyses, each incremental increase in NLR was associated with increases in all-cause mortality (crude HR: 1.028, 95% CI: 1.013–1.042, P < 0.001) and cardiac-related mortality (crude HR: 1.025, 95% CI: 1.004–1.047, P = 0.021) but not MACE (crude HR: 1.014, 95% CI: 0.990–1.038, P = 0.267) or MALE (crude HR: 1.012, 95% CI: 0.987–1.038, P = 0.346). Incremental increases in NLR were not associated with the secondary outcomes (crude HR: 1.012, 95% CI: 0.965–1.061, P = 0.626). We demonstrate there are associations between NLR and both variables and study outcomes in Table 2. With regard for mortality and MACE, other factors associated with the study outcomes included baseline heart rate and the presence of acute limb ischemia, whereas the Rutherford stage was associated with MALE but not all-cause and cardiac-related mortality and MACE.

After we adjusted for confounders in the multivariate logistic regression analyses, each incremental increase in NLR was significantly associated with increased all-cause mortality

**Table 1. Baseline and procedural characteristics and laboratory data for patients with critical limb ischemia.**

| | NLR < 8 | | NLR ≥ 8 | | P |
|---|---|---|---|---|---|
| | N = 138 | | N = 57 | | |
| Age (years) | 73.6 | (11.8) | 75.1 | (10.9) | 0.393 |
| Male sex | 70 | 50.7% | 31 | 54.4% | 0.753 |
| Body mass index (kg/m$^2$) | 23.6 | (4.0) | 23.0 | (4.5) | 0.358 |
| Heart rate at baseline (beats per minute) | 86.1 | (15.6) | 93.3 | (21.7) | 0.025 |
| Systolic BP at baseline (mm Hg) | 148.6 | (31.0) | 144.0 | (33.1) | 0.407 |
| Diastolic BP at baseline (mm Hg) | 74.7 | (13.0) | 76.3 | (17.7) | 0.531 |
| Current/past smoker | 35 | 25.4% | 11 | 19.3% | 0.459 |
| Alcohol intake | 38 | 27.5% | 17 | 29.8% | 0.730 |
| Family history of premature CAD | 2 | 1.4% | 0 | 0% | 1.000 |
| History of hypertension | 97 | 70.3% | 32 | 56.1% | 0.068 |
| History of diabetes mellitus | 97 | 70.3% | 37 | 64.9% | 0.499 |
| History of insulin use | 20 | 14.5% | 5 | 8.8% | 0.350 |
| History of dyslipidemia | 22 | 15.9% | 11 | 19.3% | 0.675 |
| Normal kidney function | 89 | 64.5% | 34 | 59.6% | 0.150 |
| Chronic kidney disease | 26 | 18.4% | 7 | 12.3% | |
| End-stage renal disease | 23 | 16.7% | 16 | 28.1% | |
| History of CAD | 52 | 37.7% | 23 | 40.4% | 0.748 |
| History of myocardial infarction | 11 | 8.0% | 2 | 3.5% | 0.355 |
| History of atrial fibrillation | 21 | 15.2% | 5 | 8.8% | 0.258 |
| History of chronic heart failure | 29 | 21.0% | 13 | 22.8% | 0.920 |
| NYHA class I | 6 | 4.3% | 4 | 7.0% | |
| NYHA class II | 8 | 5.8% | 4 | 7.0% | |
| NYHA class III | 11 | 8.0% | 4 | 7.0% | |
| NYHA class IV | 4 | 2.9% | 1 | 1.8% | |
| History of carotid artery stenosis | 3 | 2.2% | 0 | 0% | 0.557 |
| History of ischemic stroke | 24 | 17.4% | 10 | 17.5% | 1.000 |
| Ongoing cancer | 10 | 7.1% | 2 | 3.5% | 0.514 |
| History of amputation | | | | | 0.991 |
| Above-knee amputation | 4 | 2.9% | 2 | 3.5% | |
| Below-knee amputation | 2 | 1.4% | 1 | 1.8% | |
| Forefoot amputation | 3 | 2.2% | 1 | 1.8% | |
| Presented with acute ischemic limb | 11 | 8.0% | 17 | 29.8% | <0.001 |
| Rutherford classification | | | | | 0.005 |
| Class IV | 44 | 31.9% | 9 | 15.8% | |
| Class V | 89 | 64.5% | 40 | 70.2% | |
| Class VI | 5 | 3.6% | 8 | 14.0% | |
| Laboratory data | | | | | |
| Total cholesterol (mg/dL) | 159.2 | (46.8) | 148.0 | (48.8) | 0.160 |
| High-density lipoprotein cholesterol (mg/dL) | 40.9 | (18.6) | 38.2 | (19.2) | 0.639 |
| Low-density lipoprotein cholesterol (mg/dL) | 95.0 | (34.9) | 83.2 | (43.7) | 0.257 |
| Triglyceride (mg/dL) | 151.8 | (121.7) | 117.9 | (83.2) | 0.205 |
| Fasting glucose (mg/dL) | 177.7 | (95.0) | 184.9 | (126.2) | 0.663 |
| Glycosylated hemoglobin (%) | 7.4 | (1.8) | 7.6 | (2.1) | 0.684 |
| Creatinine (mg/dL) | 3.3 | (3.1) | 3.8 | (3.6) | 0.288 |
| Alanine transaminase (IU/L) | 18.5 | (10.4) | 33.6 | (35.2) | 0.003 |
| Uric acid (mg/dL) | 5.6 | (2.2) | 6.1 | (2.9) | 0.248 |

*(Continued)*

**Table 1.** (Continued)

| | NLR < 8 | | NLR ≥ 8 | | P |
|---|---|---|---|---|---|
| | N = 138 | | N = 57 | | |
| White blood cell count $10^3$/μL | 7.8 | (3.1) | 14.4 | (5.8) | <0.001 |
| Neutrophil ratio (%) | 65.3 | (12.2) | 84.9 | (5.1) | <0.001 |
| Lymphocyte ratio (%) | 20.3 | (7.8) | 5.9 | (2.5) | <0.001 |
| NLR | 3.9 | (1.9) | 18.7 | (13.5) | <0.001 |
| **Medication use at baseline** | | | | | |
| Aspirin | 49 | 35.5% | 18 | 31.6% | 0.623 |
| Cilostazol | 72 | 52.2% | 18 | 31.6% | 0.011 |
| Clopidogrel | 35 | 25.4% | 18 | 31.6% | 0.381 |
| Pentoxifylline | 8 | 5.8% | 0 | 0% | 0.108 |
| ACEI or ARB | 7 | 5.1% | 2 | 3.5% | 1.000 |
| Beta-blocker | 26 | 18.8% | 9 | 15.8% | 0.686 |
| Calcium channel blocker | 29 | 21.0% | 10 | 17.5% | 0.695 |
| Statin | 28 | 20.3% | 9 | 15.8% | 0.550 |
| Urate lowering therapy | 3 | 2.2% | 2 | 3.5% | 0.631 |
| **Procedure characteristics** | | | | | |
| **Ischemia-related artery** | | | | | |
| Iliac artery involvement | 16 | 11.6% | 7 | 12.3% | 1.000 |
| Superficial femoral artery involvement | 90 | 65.2% | 33 | 57.9% | 0.415 |
| Below the knee artery involvement | 25 | 18.1% | 10 | 17.5% | 1.000 |
| In-stent restenosis | 2 | 1.4% | 2 | 3.5% | 0.582 |

Values are expressed as numbers (standard deviation) or numbers and percentages.

CAD = coronary artery disease; NYHA = New York Heart Association; BP = blood pressure; NLR = neutrophil-to-lymphocyte ratio; ACEI = angiotensin-converting enzyme inhibitor; ARB = angiotensin receptor blocker

(adjusted HR: 1.028, 95% CI: 1.008–1.049, P = 0.007) and cardiac-related mortality (adjusted HR: 1.027, 95% CI: 0.998–1.057, P = 0.073), but there was no significant association between NLR and either MACE or MALE. NLR ≥ 8 and NLR < 8 were significantly associated with all-cause mortality (adjusted HR: 3.599, 95% CI: 1.818–7.123, P < 0.001) and cardiac-related mortality (adjusted HR: 5.286, 95% CI: 2.075–13.47, P < 0.001) but not MACE. NLR ≥ 6 and NLR < 6 were significantly associated with MALE (adjusted HR: 2.804, 95% CI: 1.292–6.088, P = 0.009) (Table 2). Kaplan-Meier curves demonstrating the incidence of the study outcomes are shown in Fig 2.

## Discussion

In the present study, we show that elevated NLR was associated with one-year all-cause and cardiac-related mortality in patients with CLI but was not associated with one-year MACE or MALE. The associations between the NLR and our study outcomes can be clinically explained by the functions of neutrophils and lymphocytes in the human immune system. The percentage of white blood cells accounted for by neutrophils increases as the immune response activates against bacterial infection in CLI patients with sepsis; a higher neutrophil ratio indicates a stronger inflammatory response and is associated with a poor prognosis in patients with LEAD [15]. Lymphocytes play a role in adaptive immunity, and a lower lymphocyte ratio was found to indicate a lower survival probability in previous studies of patients with sepsis [16]. In our study, patients with a lower lymphocytes ratio were relatively immunocompromised

**Table 2. Variables associated with study outcomes in patients with critical limb ischemia in logistic regression analyses.**

| | cHR | 95% CI | | P | aHR | 95% CI | | P | aHR | 95% CI | | P |
|---|---|---|---|---|---|---|---|---|---|---|---|---|
| **All-cause mortality at one year** | | | | | | | | | | | | |
| NLR as a continuous variable[a] | 1.028 | 1.013 | 1.042 | <0.001 | 1.028 | 1.008 | 1.049 | 0.007 | | | | |
| NLR ≥ 8 vs. < 8[a] | 5.272 | 2.972 | 9.353 | <0.001 | | | | | 3.599 | 1.818 | 7.123 | <0.001 |
| Age (years) | 1.031 | 1.004 | 1.059 | 0.023 | 1.013 | 0.983 | 1.044 | 0.400 | 1.014 | 0.983 | 1.045 | 0.393 |
| Male sex | 0.698 | 0.399 | 1.220 | 0.206 | 0.487 | 0.249 | 0.954 | 0.036 | 0.535 | 0.278 | 1.028 | 0.061 |
| Body mass index (kg/m$^2$) | 0.937 | 0.867 | 1.012 | 0.096 | 0.938 | 0.862 | 1.022 | 0.143 | 0.939 | 0.867 | 1.018 | 0.126 |
| Smoking (yes or no) | 0.979 | 0.585 | 1.637 | 0.935 | | | | | | | | |
| Hypertension (yes or no) | 0.618 | 0.353 | 1.080 | 0.091 | | | | | | | | |
| Heart rate (beats per minute) | 1.025 | 1.009 | 1.042 | 0.002 | 1.016 | 0.997 | 1.036 | 0.095 | 1.012 | 0.993 | 1.032 | 0.205 |
| Alanine transaminase (IU/L) | 1.020 | 1.012 | 1.028 | <0.001 | 1.011 | 1.001 | 1.022 | 0.045 | 1.010 | 0.999 | 1.021 | 0.063 |
| Acute ischemic limb (yes or no) | 3.607 | 1.965 | 6.620 | <0.001 | 1.773 | 0.781 | 4.023 | 0.171 | 1.173 | 0.497 | 2.773 | 0.715 |
| Rutherford classification (IV, V or VI) | 1.809 | 1.051 | 3.114 | 0.032 | 1.839 | 0.978 | 3.457 | 0.059 | 1.537 | 0.830 | 2.846 | 0.172 |
| **Cardiac-related mortality at one year** | | | | | | | | | | | | |
| NLR as a continuous variable [a] | 1.025 | 1.004 | 1.047 | 0.021 | 1.027 | 0.998 | 1.057 | 0.073 | | | | |
| NLR ≥ 8 vs. < 8 [a] | 5.924 | 2.609 | 13.450 | <0.001 | | | | | 5.286 | 2.075 | 13.47 | <0.001 |
| Age (years) | 1.031 | 0.993 | 1.071 | 0.107 | 1.015 | 0.975 | 1.056 | 0.464 | 1.015 | 0.973 | 1.059 | 0.482 |
| Male sex | 0.693 | 0.315 | 1.527 | 0.363 | 0.539 | 0.222 | 1.308 | 0.172 | 0.577 | 0.243 | 1.370 | 0.212 |
| Body mass index (kg/m$^2$) | 0.940 | 0.846 | 1.044 | 0.247 | 0.945 | 0.845 | 1.057 | 0.352 | 0.942 | 0.846 | 1.049 | 0.277 |
| Smoking (yes or no) | 1.113 | 0.561 | 2.207 | 0.759 | | | | | | | | |
| Hypertension (yes or no) | 0.612 | 0.278 | 1.349 | 0.223 | | | | | | | | |
| Heart rate (beats per minute) | 1.039 | 1.017 | 1.062 | 0.001 | 1.030 | 1.006 | 1.055 | 0.014 | 1.024 | 1.001 | 1.049 | 0.045 |
| Alanine transaminase (IU/L) | 1.019 | 1.007 | 1.031 | 0.002 | 1.009 | 0.995 | 1.024 | 0.199 | 1.008 | 0.994 | 1.022 | 0.278 |
| Acute ischemic limb (yes or no) | 3.367 | 1.401 | 8.088 | 0.007 | 1.740 | 0.589 | 5.139 | 0.316 | 1.024 | 0.327 | 3.203 | 0.968 |
| Rutherford classification (IV, V or VI) | 1.001 | 0.475 | 2.110 | 0.997 | 1.108 | 0.486 | 2.527 | 0.807 | 0.899 | 0.404 | 2.012 | 0.795 |
| **Major adverse cardiac events at one year** | | | | | | | | | | | | |
| NLR as a continuous variable [a] | 1.014 | 0.990 | 1.038 | 0.267 | 1.007 | 0.974 | 1.041 | 0.681 | | | | |
| NLR ≥ 8 vs. < 8 [a] | 2.503 | 1.290 | 4.858 | 0.007 | | | | | 1.864 | 0.858 | 4.049 | 0.116 |
| Age (years) | 1.012 | 0.981 | 1.043 | 0.459 | 0.993 | 0.960 | 1.026 | 0.665 | 0.993 | 0.960 | 1.027 | 0.694 |
| Male sex | 0.766 | 0.394 | 1.490 | 0.432 | 0.628 | 0.311 | 1.268 | 0.194 | 0.609 | 0.302 | 1.230 | 0.167 |
| Body mass index (kg/m$^2$) | 0.942 | 0.864 | 1.027 | 0.176 | 0.933 | 0.851 | 1.023 | 0.142 | 0.933 | 0.851 | 1.023 | 0.139 |
| Smoking (yes or no) | 1.472 | 0.881 | 2.458 | 0.140 | | | | | | | | |
| Hypertension (yes or no) | 0.741 | 0.377 | 1.457 | 0.385 | | | | | | | | |
| Heart rate (beats per minute) | 1.025 | 1.007 | 1.043 | 0.005 | 1.027 | 1.007 | 1.049 | 0.009 | 1.025 | 1.005 | 1.046 | 0.017 |
| Alanine transaminase (IU/L) | 1.008 | 0.997 | 1.020 | 0.147 | | | | | | | | |
| Acute ischemic limb (yes or no) | 2.381 | 1.115 | 5.085 | 0.025 | 2.324 | 0.976 | 5.531 | 0.057 | 1.799 | 0.709 | 4.566 | 0.217 |
| Rutherford classification (IV, V or VI) | 1.033 | 0.561 | 1.903 | 0.917 | 1.097 | 0.577 | 2.085 | 0.777 | 1.015 | 0.529 | 1.945 | 0.965 |
| **Major adverse limb events at one year** | | | | | | | | | | | | |
| NLR as a continuous variable [a] | 1.012 | 0.987 | 1.038 | 0.346 | 0.998 | 0.965 | 1.031 | 0.897 | | | | |
| NLR ≥ 6 vs. < 6 [a] | 3.286 | 1.601 | 6.746 | 0.001 | | | | | 2.804 | 1.292 | 6.088 | 0.009 |
| Age (years) | 0.977 | 0.950 | 1.004 | 0.097 | 0.952 | 0.921 | 0.985 | 0.004 | 0.956 | 0.924 | 0.990 | 0.012 |
| Male sex | 1.247 | 0.634 | 2.455 | 0.522 | 1.376 | 0.672 | 2.819 | 0.382 | 1.350 | 0.655 | 2.783 | 0.416 |
| Body mass index (kg/m$^2$) | 0.955 | 0.876 | 1.040 | 0.288 | 0.917 | 0.836 | 1.007 | 0.069 | 0.924 | 0.840 | 1.016 | 0.104 |
| Smoking (yes or no) | 1.052 | 0.570 | 1.941 | 0.872 | | | | | | | | |
| Hypertension (yes or no) | 0.806 | 0.404 | 1.610 | 0.541 | | | | | | | | |
| Heart rate (beats per minute) | 0.998 | 0.979 | 1.017 | 0.816 | | | | | | | | |
| Alanine transaminase (IU/L) | 1.001 | 0.987 | 1.016 | 0.852 | | | | | | | | |
| Acute ischemic limb (yes or no) | 0.830 | 0.292 | 2.355 | 0.726 | 0.725 | 0.210 | 2.498 | 0.610 | 0.506 | 0.143 | 1.786 | 0.290 |

*(Continued)*

**Table 2.** (Continued)

| | cHR | 95% CI | | P | aHR | 95% CI | | P | aHR | 95% CI | | P |
|---|---|---|---|---|---|---|---|---|---|---|---|---|
| Rutherford classification (IV, V or VI) | 3.938 | 2.101 | 7.382 | <0.001 | 5.346 | 2.531 | 11.291 | <0.001 | 4.429 | 2.087 | 9.400 | <0.001 |
| **In-hospital mortality** | | | | | | | | | | | | |
| NLR as a continuous variable [a] | 1.012 | 0.965 | 1.061 | 0.626 | | | | | | | | |
| NLR ≥ 5 vs. < 5 [a] | 0.649 | 0.174 | 2.418 | 0.519 | | | | | | | | |
| Age (years) | 0.992 | 0.947 | 1.040 | 0.752 | | | | | | | | |
| Male sex | 0.421 | 0.147 | 1.207 | 0.108 | | | | | | | | |
| Body mass index (kg/m$^2$) | 1.050 | 0.898 | 1.227 | 0.542 | | | | | | | | |
| Smoking (yes or no) | 1.308 | 0.440 | 3.892 | 0.629 | | | | | | | | |
| Hypertension (yes or no) | 1.074 | 0.404 | 2.861 | 0.886 | | | | | | | | |
| Heart rate (beats per minute) | 0.999 | 0.981 | 1.018 | 0.954 | | | | | | | | |
| Alanine transaminase (IU/L) | 0.997 | 0.985 | 1.010 | 0.660 | | | | | | | | |
| Acute ischemic limb (yes or no) | 0.664 | 0.253 | 1.746 | 0.407 | | | | | | | | |
| Rutherford stages (IV, V or VI) | 1.374 | 0.670 | 2.816 | 0.386 | | | | | | | | |
| White blood count | 1.000 | 1.000 | 1.000 | 0.134 | | | | | | | | |
| Neutrophils | 0.986 | 0.938 | 1.036 | 0.574 | | | | | | | | |
| Lymphocytes | 1.042 | 0.958 | 1.134 | 0.339 | | | | | | | | |

[a] NLR as a continuous variable and NLR as a binary variable were not simultaneously adjusted in multivariate logistic regression analyses. cHR = crude hazard ratio; aHR = adjusted hazard ratio; CI: confidence interval.

and predisposed to all-cause mortality resulting from CLI-induced sepsis. In addition, experimental evidence has shown that lymphocytes activate and modify ischemia-reperfusion injury and wound healing [17]. Besides, the patients with severe physiologic stresses such as cardiogenic shock or heart failure would have elevated NLR. As mentioned above, the NLR is an inflammatory biomarker and a biomarker of poor wound healing in patients with CLI that is associated with an increased risk of all-cause mortality. Statistically, the neutrophil ratio was positively associated and the lymphocyte ratio was inversely associated with all-cause mortality. As it considers both the neutrophil and the lymphocyte ratios with regard for all-cause mortality, the NLR is a more powerful predictor of all-cause mortality than using wither the neutrophil or lymphocyte ratio alone. The CLI patients had high mortality, and an NLR ratio can be used as a simple predictor of all-cause mortality independent of the Rutherford stages and the presence of acute limb ischemia. Therefore, we think that the patients with NLR < 8 should receive aggressive endovascular treatment, even catheter-directed thrombolysis, to improve survival probability and limb salvage. On the contrary, the patient with NLR ≥ 8 undergoing percutaneous transluminal angioplasty still had high mortality due to inflammation and sepsis, although they received advanced antibiotic treatment and wound debridement. In this condition, surgical intervention such as bypass surgery or amputation might be the other therapeutic choice for patients with NLR ≥ 8.

A strength of our study is that we report comprehensive outcomes, including MACE. All-cause mortality has been reported in many studies [12, 18–21], cardiac morality in only a few studies [12, 19], and amputation in only a few studies [12, 18, 20, 21]; however, no study has previously reported MACE and MALE in patients with CLI or stable LEAD with regard for the NLR. The best cut-off for NLR, as determined by receiver operating characteristic curves, was eight in the present study, but this cut-off for NLR is not consistent with previous studies, in which it has also not been universal and ranges from three to five [12, 13, 18–21]. Spark et al. [12] previously reported that NLR ≥ 5.25 vs. NLR < 5.25 was associated with all-cause

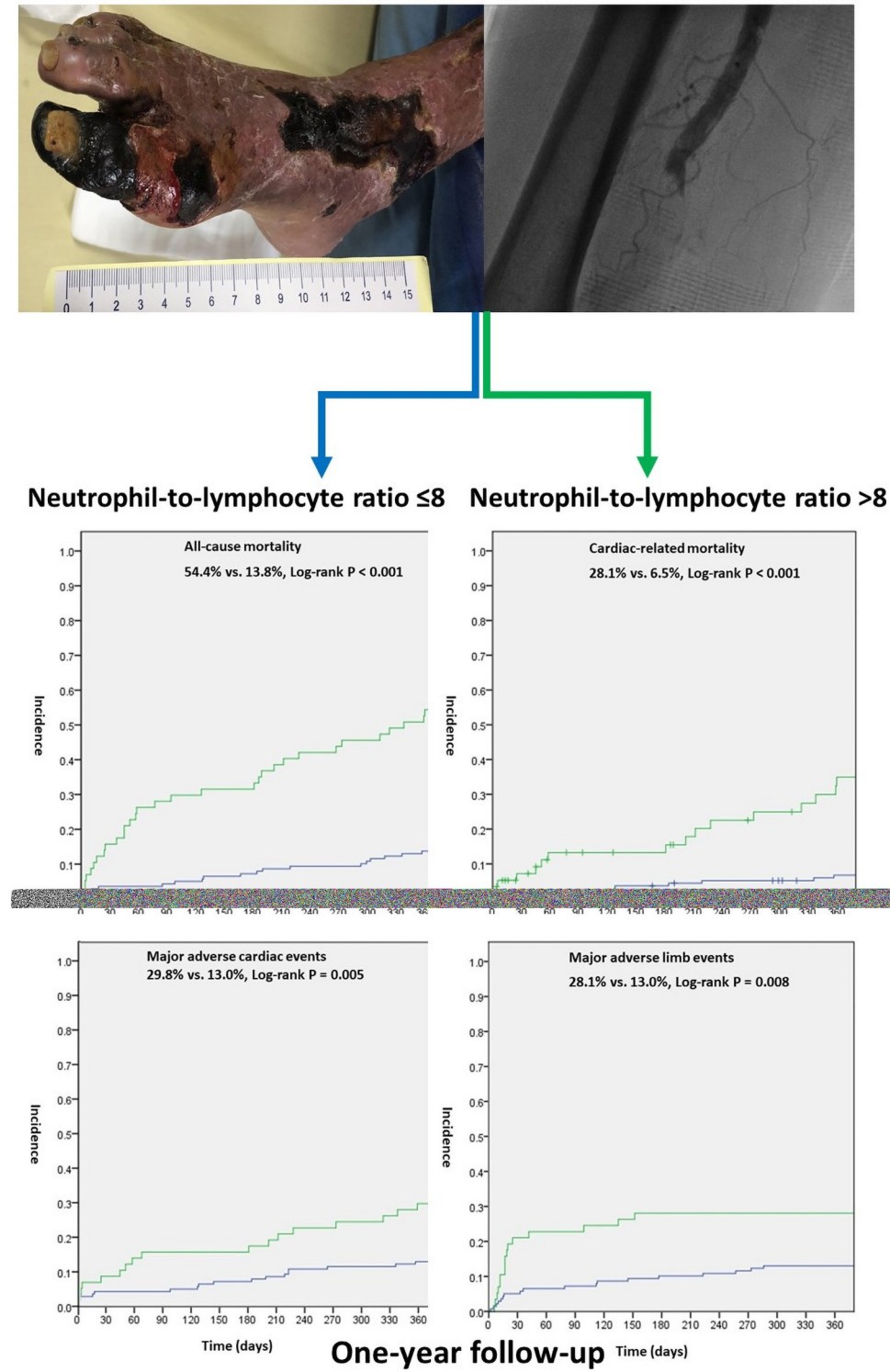

**Fig 2. Kaplan-Meier curves show that the risks of one-year all-cause (A) and cardiac-related (B) mortality, MACE (C), MALE (D), and in-hospital mortality were significantly higher in CLI patients with NLR $\geq$ 8 than in those with NLR $<$ 8.**

mortality, and an NLR cut-off of 5.25 was further validated in the study by Chan et al. [19] Notably, the study by Spark et al. [12] enrolled patients with Rutherford stages IV and V but not Rutherford stage VI, and Chan et al. [19] enrolled only one patient (less than 1%) with Rutherford stage VI. In the study by Erturk et al. [13], the NLR cut-off was even lower, at three, because more than sixty percent of the patients were at Rutherford stages III and IV. The major differences between our and other studies are related to the study populations; compared to previous investigators who enrolled patients with peripheral artery diseases and weaker inflammatory responses, we enrolled CLI patients with stronger inflammatory responses [13, 20]. In other words, the neutrophil ratio in our patients indicates a stronger inflammatory response, whereas patients in the other study had weaker inflammation [12]. In studies enrolling patients with either CLI or stable LEAD [20], the cut-off for NLR may be reduced by patients with stable LEAD, and the resulting lower NLR threshold may be unable to adequately stratify CLI patients with higher cardiovascular risks. We conducted the present study by enrolling only CLI patients while excluding patients with stable LEAD to make our study population homogeneous, and we therefore suggest that an NLR cut-off of eight could be used in CLI patients to evaluate the risk of all-cause mortality in future studies.

The NLR value increased with higher Rutherford stages in the present study, consistent with a previous study by Bath et al. [20]. Another study also showed that disease severity and the values of high-sensitivity C-reactive protein significantly increased across tertiles of NLR in patients with lower limb arteriosclerosis obliterans [21]. NLR, as a biomarker of the inflammatory response and disease severity, has been proposed to contribute to mortality in CLI patients, but NLR as a continuous variable was not associated with MALE in the present study. The NLR was also associated with the complexity of CLI-related arteries, and atherothrombosis has also been shown to be an additional risk factor for MALE in patients with CLI [22, 23]. Because CLI patients had increased platelet activation at baseline than was found in healthy subjects [24], the platelet to lymphocyte ratio may be a better predictor of incident MALEs in CLI patients [18].

A weakness of our study is that we found that the prescription of medication for CLI patients was relatively low at baseline, including the use of statin and angiotensin-converting enzyme inhibitors or angiotensin receptor blockers. Additionally, the all-cause mortality rate was as high at approximately 25.4%, and the rate of MALE was 17.3%. A meta-analysis that included thirteen studies showed that the rates of all-cause mortality and MALE were both 22% in the natural history of untreated CLI patients [25]. Another retrospective cohort study showed that four-year mortality was significantly higher among the subgroups at 37.7%, 52.2%, and 63.5% in Rutherford stages IV, V, and VI, respectively; while the four-year rates of amputation were 12.1%, 35.3%, and 67.3% for Rutherford classification IV, V, and VI, respectively [26]. The REACH registry showed that statin use was significantly associated with a decreased risk of mortality in patients with artherothrombosis [23]. Although the present study was an all-comer study, the guideline-direct medical therapy to patients at the stage of symptomatic or asymptomatic LEAD could be enhanced to improve clinical outcomes before these patients present with CLI. Given that statins have anti-inflammatory properties and pleiotropic effects on reducing mortality and MALE, future studies should investigate the effect of statin intensity on changes in NLR. Second, the prescription rate of anti-platelet agents was sixty percent at baseline in the CLI patients, while the rate at discharge was 97%. The contemporary guidelines suggest that an anti-platelet, such as aspirin or clopidogrel, is strongly recommended in patients with LEAD [27]. Although we prescribed anti-platelet agents to CLI patients according to the affordable rule of the National Health Insurance program, mortality and MALE remained high. The high incidence of mortality and MALE in the CLI patients in the present and previous studies indicates that the use of only anti-platelet

agents might not be good enough to prevent future cardiovascular events. A previous double randomized controlled trial enrolled patients with atherosclerotic cardiovascular diseases and showed that compared to aspirin alone, low-dose rivaroxaban plus aspirin reduced MALE with the trade-off of bleeding complications [28]. The VOYAGER study randomized patients with LEAD undergoing revascularization to a low-dose rivaroxaban plus aspirin group or an aspirin alone group and found that randomization to low-dose rivaroxaban plus aspirin was associated with the significantly lower incidence of MACE and MALE than randomization to aspirin alone [28]; the trade-off of an increased risk of major bleeding was also reported in the VOYAGER study. Regarding the net clinical benefit of reduced MALE and increased bleeding complications, we propose that NLR $\geq$ 8 may be useful for stratifying CLI patients with a high risk of mortality who may benefit from additional low-dose rivaroxaban in addition to aspirin. Another limitation is that we investigated the association between NLR and several outcomes, but the best cut-off of NLR was not consistent across all study outcomes in our own study. NLR $\geq$ 8 vs. < 8 was associated with an increased risk of mortality but not MALE, whereas NLR $\geq$ 6 vs. NLR < 6 was associated with the incidence of MALE. Given that it would be difficult to apply multiple cut-off values in a clinical setting and difficult to validate them externally in other studies, we suggest that a cut-off of NLR $\geq$ 8 vs. < 8 may be adequate for identifying CLI patients at a risk of high mortality. Other predictors, such as the platelet-lymphocyte ratio, should also be investigated regarding the incidence of MALE in CLI patients. In addition, a systematic review and meta-analysis could be performed to determine the best cut-off value for NLR among patients with LEAD with various risks.

We previously reported that elevated serum uric acid was associated with the prevalence of metabolic syndrome, left ventricular hypertrophy, left ventricular diastolic dysfunction in various study populations such as apparently healthy individuals, patients with cardiometabolic abnormalities, and patients with ST-segment elevation myocardial infarction [29–32]. However, we did not find a significant association between hyperuricemia and MACE and MALE in the present study. The possible explanation may be that serum uric acid serves as weaker inflammation, and it, therefore, did not play a role in CLI patients with stronger inflammatory responses.

## Conclusions

We show that CLI patients with a higher NLR have significantly higher incidences of all-cause and cardiac-related mortality and MALE at one year. Each increment in NLR was associated with an increased risk of all-cause and cardiac-related mortality but not MALE. NLR $\geq$ 8 vs. NLR < 8 can be used to predict mortality in CLI patients.

## Supporting information

**S1 Table. Baseline and procedural characteristics and laboratory data in patients with critical limb ischemia.** Values were expressed as number and (standard deviation) or number and percentage. CAD = coronary artery disease; NYHA = New York Heart Association; BP = blood pressure; NLR = neutrophil-lymphocyte ratio; ACEI = angiotensin-converting enzyme inhibitors; ARB = angiotensin receptor blockers.
(TIF)

**S2 Table. Baseline and procedural characteristics and laboratory data in patients with critical limb ischemia.** Values were expressed as number and (standard deviation) or number and percentage. CAD = coronary artery disease; NYHA = New York Heart Association; BP = blood pressure; NLR = neutrophil-lymphocyte ratio; ACEI = angiotensin-converting

enzyme inhibitors; ARB = angiotensin receptor blockers.
(TIF)

## Author Contributions

**Conceptualization:** Min-I. Su.

**Data curation:** Min-I. Su.

**Formal analysis:** Cheng-Wei Liu.

**Investigation:** Cheng-Wei Liu.

**Methodology:** Cheng-Wei Liu.

**Writing – original draft:** Cheng-Wei Liu.

**Writing – review & editing:** Cheng-Wei Liu.

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
