## [Decision Letter · Decision Letter 0]

29 Apr 2021

PONE-D-21-11099

Neutrophil-to-lymphocyte ratio associated with an increased risk of mortality in patients with critical limb ischemia

PLOS ONE

Dear Dr.  Liu,

Thank you for submitting your manuscript to PLOS ONE. After careful consideration, we feel that it has merit but does not fully meet PLOS ONE’s publication criteria as it currently stands. Therefore, we invite you to submit a revised version of the manuscript that addresses the points raised during the review process.

Editor's comments:

Overall well written manuscript, 5 years data, overall low sample size due to CLI. Please address following comments:

- low extremity artery disease (LEAD)- would be lower extremity not low extremity.

-Please expand the clinical utility of this results, How we prevent mortality in CLI patients after checking NLR?

- although the strengths and limitations have been mentioned I suggest to write all limitations in one paragraph

Please submit your revised manuscript by May 30th.  If you will need more time than this to complete your revisions, please reply to this message or contact the journal office at plosone@plos.org. Please include the following items when submitting your revised manuscript:

We look forward to receiving your revised manuscript.

Kind regards,

Timir Paul

Academic Editor

PLOS ONE

Journal Requirements:

2. Please upload a copy of Supporting Information Table S1 which you refer to in your text on page 8.

Reviewers' comments:

Reviewer's Responses to Questions

**Comments to the Author**

1. Is the manuscript technically sound, and do the data support the conclusions?

Reviewer #1: Yes

2. Has the statistical analysis been performed appropriately and rigorously? 

Reviewer #1: N/A

3. Have the authors made all data underlying the findings in their manuscript fully available?

Reviewer #1: Yes

4. Is the manuscript presented in an intelligible fashion and written in standard English?

Reviewer #1: Yes

5. Review Comments to the Author

Reviewer #1: Very nice article and focus on important aspect of cardiology. I would like to know more about the NLR measurement, is it the first CBC or is it the hospitalization count average, pre and post intervention, follow up NLR ?... did the authors depend on just the initial NLR for their study. Also, would like to know if there was any other medical problems that cofound or affect the ratio ?

6. PLOS authors have the option to publish the peer review history of their article (what does this mean?). If published, this will include your full peer review and any attached files.

Reviewer #1: No

---

## [Author Response · Author response to Decision Letter 0]

5 May 2021

Editor's comments:

Overall well written manuscript, 5 years data, overall low sample size due to CLI. Please address following comments:

- low extremity artery disease (LEAD)- would be lower extremity not low extremity.

Response to the editor:

We replaced "low extremity" with "lower extremity" (page 4, line 60.) Thanks for the editor's comment.

-Please expand the clinical utility of this results, How we prevent mortality in CLI patients after checking NLR?

Response to the editor:

The CLI patients had high mortality, and an NLR ratio can be used as a simple predictor of all-cause mortality independent of the Rutherford stages and the presence of acute limb ischemia. Therefore, we think that the patients with NLR < 8 should receive aggressive endovascular treatment, even catheter-directed thrombolysis, to improve survival probability and limb salvage. On the contrary, the patient with NLR ≥ 8 undergoing percutaneous transluminal angioplasty still had high mortality due to inflammation and sepsis, although they received advanced antibiotic treatment and wound debridement. In this condition, surgical intervention such as bypass surgery or amputation might be the other therapeutic choice for patients with NLR ≥ 8. We added this paragraph in the first paragraph of the discussions. (page 20, line 198-206)

- although the strengths and limitations have been mentioned I suggest to write all limitations in one paragraph

Response to the editor:

We combined the limitations into one paragraph as the editor's comment, including "A weakness of our study……" (page 22 line 240) and "another limitation……" (page 24 line 272).

Reviewer #1: 

(1)Very nice article and focus on important aspect of cardiology. I would like to know more about the NLR measurement, is it the first CBC or is it the hospitalization count average, pre and post intervention, follow up NLR ? 

Response to reviewer #1: 

Thanks for the review's appreciation. We measured complete blood count and white blood cell differential at our emergency department before the patients received percutaneous transluminal angioplasty.

 We added a sentence in the second paragraph of the method section. "We measured complete blood count and white blood cell differential at our emergency department before the patients received percutaneous transluminal angioplasty." (page 5 line80-81)

Reviewer #1:

(2) ... did the authors depend on just the initial NLR for their study. Also, would like to know if there was any other medical problems that cofound or affect the ratio ?

Response to reviewer #1:

As we described from page 19, line 183 to page 20, line 190, "The percentage of white blood cells accounted for by neutrophils increases as the immune response activates against bacterial infection in CLI patients with sepsis; a higher neutrophil ratio indicates a stronger inflammatory response and is associated with a poor prognosis in patients with LEAD [15]. Lymphocytes play a role in adaptive immunity, and a lower lymphocyte ratio was found to indicate a lower survival probability in previous studies of patients with sepsis [16]. In our study, patients with a lower lymphocytes ratio were relatively immunocompromised and predisposed to all-cause mortality resulting from CLI-induced sepsis." Therefore, inflammatory response induced by both sepsis and atherosclerotic plaques can increase neutrophil counts. Besides, the patients with severe stresses such as cardiogenic shock or heart failure would have elevated NLR.

 We added a sentence in the first paragraph of the discussions. "The patients with severe physiologic stresses such as cardiogenic shock or heart failure would have elevated NLR." (page 20 line 191-193)

---

## [Editor Report · Decision Letter 1]

10 May 2021

Neutrophil-to-lymphocyte ratio associated with an increased risk of mortality in patients with critical limb ischemia

PONE-D-21-11099R1

Dear Dr. Liu,

We’re pleased to inform you that your manuscript has been judged scientifically suitable for publication and will be formally accepted for publication once it meets all outstanding technical requirements.

Kind regards,

Timir Paul

Academic Editor

PLOS ONE

Additional Editor Comments (optional):

All comments were appropriately addressed. 
---

## [Editor Report · Acceptance letter]

17 May 2021

PONE-D-21-11099R1 

Neutrophil-to-lymphocyte ratio associated with an increased risk of mortality in patients with critical limb ischemia 

Dear Dr. Liu:

I'm pleased to inform you that your manuscript has been deemed suitable for publication in PLOS ONE. Congratulations! Your manuscript is now with our production department. 

Kind regards, 

on behalf of

Dr. Timir Paul 

Academic Editor

PLOS ONE